# Matrix Gla protein polymorphism rs1800801 associates with recurrence of ischemic stroke

Philipp Hendrix[1,2], Nelson Sofoluke[1], Matthew Adams[3], Saran Kunaprayoon[3], Ramin Zand[4], Amy N. Kolinovsky[5], Thomas N. Person[5], Mudit Gupta[5], Oded Goren[2], H. Lester Kirchner[6], Clemens M. Schirmer[1], Natalia S. Rost[7], James E. Faber[8], Christoph J. Griessenauer[1,9]*

1 Department of Neurosurgery, Geisinger, Danville, PA, United States of America, 2 Department of Neurosurgery, Saarland University Medical Center and Saarland University Faculty of Medicine, Homburg/Saar, Germany, 3 Geisinger Commonwealth School of Medicine, Scranton, PA, United States of America, 4 Department of Neurology, Geisinger, Danville, PA, United States of America, 5 Geisinger Health System Phenomic Analytics and Clinical Data Core, Danville, PA, United States of America, 6 Biomedical and Translational Informatics, Geisinger Health System, Danville, PA, United States of America, 7 Department of Neurology, Massachusetts General Hospital, Harvard Medical School, Boston, MA, United States of America, 8 Department of Cell Biology and Physiology, University of North Carolina, Chapel Hill, NC, United States of America, 9 Research Institute of Neurointervention, Paracelsus Medical University, Salzburg, Austria

* christoph.griessenauer@gmail.com

**Data Availability Statement:** Geisinger is not the sole proprietor of the data used in this manuscript. The Geisinger MyCode initiative is an academia-industry collaboration between Geisinger and

## Abstract

The MGP single nucleotide polymorphism (SNP) rs1800801 has previously been associated with recurrent ischemic stroke in a Spanish cohort. Here, we tested for association of this SNP with ischemic stroke recurrence in a North American Caucasian cohort. Acute ischemic stroke patients admitted between 10/2009 and 12/2016 at three hospitals within a large healthcare system in the northeastern United States that were enrolled in a healthcare system-wide exome sequencing program were retrospectively reviewed. Patients with recurrent stroke within 1 year after index event were compared to those without recurrence. Of 9,348 suspected acute ischemic strokes admitted between 10/2009 and 12/2016, 1,727 (18.5%) enrolled in the exome-sequencing program. Among those, 1,068 patients had exome sequencing completed and were eligible for inclusion. Recurrent stroke within the first year of stroke was observed in 79 patients (7.4%). In multivariable analysis, stroke prior to the index stroke (OR 9.694, 95% CI 5.793–16.224, p $\leq$ 0.001), pro-coagulant status (OR = 3.563, 95% CI 1.504–8.443, p = 0.004) and the AA genotype of SNP rs1800801 (OR = 2.408, 95% CI 1.079–4.389, p = 0.004) were independently associated with recurrent stroke within the first year. The AA genotype of the MGP SNP rs1800801 is associated with recurrence within the first year after ischemic stroke in North American Caucasians. Study of stroke subtypes and additional populations will be required to determine if incorporation of allelic status at this SNP into current risk scores improves prediction of recurrent ischemic stroke.

Regeneron, Geisinger's industry partner. The agreement between Geisinger and Regeneron states that summary statistics can be shared such as in this manuscript. Regeneron reviewed and approved this submission. Any data sharing on the patient level (i.e. individual patient SNPs) requires the execution of a data sharing agreement between the requestor and Geisinger that is also aligned with Geisinger-Regeneron data sharing terms. For such requests, research contracts at Geisinger and Geisinger's Institutional Review Board have to be contacted. Please direct requests to irb@geisinger. edu. Once appropriate data use agreements are executed, others with be able to access the data in the same manner. There are no special privileges others will not be able to get access to after appropriate data use agreements are executed.

**Funding:** Geisinger Health provided support in the form of salaries for certain authors [N.S., R.Z., A.N. K., T.N.P., M.G., O.G., H.L.K., C.M.S., C.J.G.], but did not have any additional role in the study design, data collection and analysis, decision to publish, or preparation of the manuscript. The specific roles of these authors are articulated in the 'author contributions' section.

**Competing interests:** The affiliation Geisinger Health does not alter our adherence to PLOS ONE policies on sharing data and materials. We also declare that there are no relevant declarations relating to consultancy, patents, products in development, or marketed products.

## Introduction

Annually, about 795,000 Americans suffer from stroke, and almost one quarter, 185,000, of these strokes are recurrent strokes [1]. Although several prediction models for recurrent stroke have been reported [2–6], their certainty is moderate at best [7]. Recent data on inclusion of genetic markers in prediction models of stroke has been mixed. Achterberg and colleagues did not find an additional value of genetic information in predicting recurrence of vascular events including stroke after cerebral ischemia [8]. In the Genotyping Recurrence Risk of Stroke (GRECOS) project, however, Fernándes-Cadenas et al. found an association of the single nucleotide polymorphism (SNP) rs1800801 in the matrix carboxyglutamatic acid Gla protein (MGP) gene with first-year recurrent ischemic stroke in Spanish Caucasians [9]. MGP is an extracellular matrix protein involved in the inhibition of calcification of arteries and cartilage [10]. Recently, a meta-analysis highlighted the SNP rs1800801 (12-15038788C-T, G>A) in Caucasians to increase the risk for vascular calcification and atherosclerotic disease [11]. Here, we investigated the impact of rs18008001 on the risk for first-year recurrent stroke in a North American Caucasian population.

## Materials and methods

### Study design

Caucasian acute ischemic stroke patients admitted between October 2009 and December 2016 at three hospitals that are part of a large healthcare system in the northeastern United States were retrospectively reviewed. Patients were identified through the American Heart Association "Get With The Guidelines® Stroke" center database and cross-checked with individuals enrolled in a healthcare system-wide bio-banking exome sequencing program to link genetic samples and electronic health records data called MyCode. The institutional phenomic analytics and clinical data core performed electronic health record (EHR) data extraction from the clinical documentation improvement specialist (CDIS) and various disparate data sources with subsequent manual chart review (N.S., M.A., S.K.) to verify the accuracy of diagnosis of acute ischemic stroke and document subsequent clinical management. This study was carried out in accordance with the recommendations of the Geisinger Institutional Review Board with written informed consent from all subjects. All subjects gave written informed consent in accordance with the Declaration of Helsinki. The protocol was approved by the Geisinger Institutional Review Board (IRB#: 2017–0521).

### MyCode enrollment process and whole exome sequencing

The MyCode Community Health Initiative was established in 2007 as a discovery research initiative with more than 200,000 currently consented subjects [12]. Recruitment occurs in primary care and specialty clinics throughout the health system without regard to underlying diseases. The health system established research collaboration with Regeneron Genetics Center (RGC) that includes conducting whole-exome sequencing in MyCode participants and linking sequence data to EHR data. Whole-exome sequencing is performed at RGC as previously described [13]. Exome-sequencing data were available for 92,455 individuals at the commencement of this study. The MGP SNP rs1800801 genotype was obtained in eligible stroke patients. The minor allele frequency (MAF) of rs1800801, which is a single nucleotide G>A variant in the 5'UTR, is 0.31 for the A allele [14].

### Patient characteristics, clinical variables, and outcome measures

The diagnosis of acute ischemic stroke was based on the neurologic examination and confirmed by CT-scan or magnetic resonance imaging (MRI). Ischemic strokes were allocated to

subtypes of ischemic stroke according to TOAST criteria [15]. Clinical and past-medical history variables were collected in a non-blinded fashion. Functional outcome was assessed using the modified Rankin Scale (mRS) with mRS 0–2 representing a favorable and mRS 3–6 representing an unfavorable functional outcome (mRS 6 death).

### First-year recurrent ischemic stroke and non-recurrent ischemic strokes

Patients who experienced an acute ischemic stroke between 10/2009 and 12/2016 were screened for first-year recurrent stroke via chart review of the health system database. Patients were categorized as (1) recurrent stroke within the first year of index stroke versus (2) no recurrent stroke. Those patients who experienced a recurrent stroke beyond the first year were excluded from this analysis.

### Statistical analysis

Continuous variables are presented as mean ± standard deviation and categorical variables are presented as frequency and percent. Univariable analyses were carried out using binary logistic regression, Chi-square and Fisher's exact tests, as appropriate. Post hoc testing for crosstabs exceeding 2x2 dimension was performed calculating adjusted standardized residuals (z-scores) and thereof p-values. Significance for crosstabs was eventually evaluated after Bonferroni correction. Multivariable analysis was performed by integrating variables with a possible association with first-year recurrent stroke (p < 0.15). Stepwise backward elimination was performed, p-values of < 0.05 were considered statistically significant. Discrimination of predictive models was performed using the area under the receiver operating characteristic curve (AUC). Statistical analysis was performed using IBM SPSS version 22. Analysis was conducted using MedCalc software.

## Results

### Patient characteristics

A total of 9,348 suspected acute ischemic strokes were admitted to one of three health system hospitals between 10/2009 and 12/2016, with 1,727 (18.5%) enrolled in the exome sequencing program. Among those, 1,068 MyCode patients had exome sequencing data available at the commencement of this study and were eligible for inclusion. The one-year incidence rate of recurrent stroke was 79/1,068 (7.4%) in our study population. Mean age of patients with and without recurrent stroke was 66.9 and 68.2 years, respectively. Functional outcome at discharge was favorable (mRS 0–2) in 28/79 (35.4%) recurrent stroke patients and 352/985 (35.7%) non-recurrent stroke patients. In-hospital mortality was 1.3% (1/79) in recurrent stroke patients and 2.4% (24/985) in non-recurrent stroke patients. At 90 days follow-up, 38/77 (49.4%) recurrent stroke patients had a favorable functional outcome, and 6/79 (7.8%) recurrent stroke patients were deceased (Table 1).

### Association of matrix Gla protein polymorphism, clinical variables and recurrent stroke

The MGP SNP rs1800801 was in Hardy-Weinberg equilibrium in patients with and without first-year recurrent stroke (p = 0.928 and p = 0.424, respectively). Genotype distributions between both groups differed significantly (p = 0.011). Linear-by-linear association demonstrated a significant association of the T allele (polymorphism allele) with the rate of recurrent strokes (p = 0.004) (Table 2).

**Table 1. Patient characteristics.**

| | Recurrent stroke within first year | No recurrent stroke | Statistics OR (95% CI) / p-value |
|---|---|---|---|
| Patients (N) | 79/1068 (7.4%) | 989/1068 (92.6%) | |
| Age in years [mean ± SD] | 66.9 ± 14.5 | 68.2 ± 13.2 | 0.387 |
| Sex | | | |
| Female | 44/79 (55.7%) | 483/989 (48.8%) | 1.317 (0.830–2.089) |
| Male | 35/79 (44.3%) | 506/989 (51.2%) | |
| TOAST subtypes of stroke | | | ≤ 0.001 |
| Cardioembolism | 24/79 (30.4%) | 313/989 (31.6%) | |
| Large-artery atherosclerosis | 25/79 (31.6%) | 222/989 (22.4%) | |
| Small vessel occlusion | 5/79 (6.3%) | 168/989 (17.0%) | |
| Stroke of undetermined etiology | 17/79 (21.5%) | 263/989 (26.6%) | |
| Stroke of other determined etiology | 8/79 (10.1%) | 23/989 (2.3%) | |
| History of hypertension | 67/79 (84.8%) | 784/989 (79.3%) | 1.460 (0.775–2.750) |
| Diabetes mellitus II | 35/79 (44.3%) | 392/989 (39.6%) | 1.211 (0.763–1.923) |
| Dyslipidemia | 64/79 (81.0%) | 704/989 (71.2%) | 1.727 (0.968–3.081) |
| Smoking status$^{\S}$ | | | |
| Current | 20/79 (25.3%) | 205/967 (21.2%) | 1.260 (0.742–2.141) |
| Former | 25/79 (31.6%) | 390/967 (40.3%) | |
| Never | 34/79 (43.0%) | 372/967 (38.5%) | |
| Alcohol consumption$^{\$}$ | 25/79 (31.6%) | 344/936 (36.8%) | 0.797 (0.487–1.304) |
| BMI ≥ 25 | 63/79 (79.7%) | 786/989 (79.5%) | 1.017 (0.575–1.798) |
| Peripheral vascular disease | 13/79 (16.5%) | 107/989 (10.8%) | 1.624 (0.867–3.041) |
| Coronary artery disease | 33/79 (41.8%) | 301/989 (30.4%) | 1.640 (1.028–2.616) |
| Atrial fibrillation | 18/79 (22.8%) | 223/989 (22.5%) | 1.014 (0.587–1.751) |
| Prior stroke | 35/79 (44.3%) | 75/989 (7.6%) | 9.694 (5.865–16.022) |
| Carotid stenosis | 28/79 (35.4%) | 384/989 (38.8%) | 0.865 (0.536–1.396) |
| Intracranial arteriosclerosis | 32/79 (40.5%) | 333/989 (33.7%) | 1.341 (0.840–2.142) |
| Anemia | 13/79 (16.5%) | 145/989 (14.7%) | 1.146 (0.617–2.132) |
| Pro-coagulant coagulation disorder | 9/79 (11.4%) | 31/989 (3.1%) | 3.973 (1.820–8.674) |
| COPD | 9/79 (11.4%) | 120/989 (12.1%) | 0.931 (0.453–1.912) |
| Sleep apnea | 11/79 (13.9%) | 104/989 (10.5%) | 1.377 (0.705–2.686) |
| Family history | 24/79 (30.4%) | 207/989 (20.9%) | 1.648 (0.996–2.727) |
| Home medication | | | |
| Anti-platelet | 53/79 (67.1%) | 291/989 (29.4%) | 4.878 (3.003–8.000) |
| Anticoagulation | 14/79 (17.7%) | 65/989 (6.6%) | 3.062 (1.631–5.748) |
| Statins | 55/79 (69.6%) | 296/989 (29.9%) | 5.365 (3.259–8.832) |
| ACE-/AT$_1$-inhibitor | 30/79 (38.0%) | 266/989 (26.9%) | 1.664 (1.034–2.678) |
| Beta blocker | 41/79 (51.9%) | 338/989 (34.2%) | 2.078 (1.311–3.293) |
| Oral anti-diabetic | 17/79 (21.5%) | 127/989 (12.8%) | 1.861 (1.055–3.284) |
| Outcome at discharge (mRS)$^{\#}$ | | | |
| 0–2 | 28/79 (35.4%) | 352/985 (35.7%) | 0.987 (0.611–1.594) |
| 6 (dead) | 1/79 (1.3%) | 24/985 (2.4%) | 0.513 (0.069–3.845) |
| Outcome after 90 days (mRS)$^{\dagger}$ | | | |
| 0–2 | 38/77 (49.4%) | 658/967 (68.0%) | 0.458 (0.287–0.730) |
| 6 (dead) | 6/77 (7.8%) | 57/967 (5.9%) | 1.349 (0.562–3.237) |

Data missing in § 22 patients, $ 53 patients, # 4 patients, † 24 patients

**Table 2. Genotype frequencies of SNP rs1800801.**

| Polymorphism | Genotype | Recurrent stroke | No recurrent stroke | P value |
|---|---|---|---|---|
| rs1800801 | GG (N = 422) | 22/422 (5.2%) | 400/422 (94.8%) | 0.011 |
| | GA (N = 506) | 39/506 (7.7%) | 467/506 (92.3%) | |
| | AA (N = 140) | 18/140 (12.9%) | 122/140 (87.1%) | |
| | HWE | 0.928 | 0.424 | |

Linear-by-linear association p = 0.004. HWE = Hardy-Weinberg equilibrium

In univariable analysis, dyslipidemia, peripheral vascular disease, coronary artery disease, prior stroke, pro-coagulant disorder, and positive family history of stroke were associated with recurrent stroke within the first year and were thus analyzed in multivariable analysis. Home medication (each of the following: anti-platelet, anticoagulation, statins, ACE-/$AT_1$-inhibitor, beta-blocker and oral anti-diabetic) was also associated with recurrent stroke within the first year. However, these variables were not integrated into the multivariable analysis as they present indirect indicator variables of cerebrovascular disease and other health conditions that were directly assessed through past medical history variables. In multivariable analysis, stroke prior to the index stroke (OR 9.694, 95% CI 5.793–16.224, p ≤ 0.001), pro-coagulant status (hypercoagulability) (OR = 3.563, 95% CI 1.504–8.443, p = 0.004) and the AA vs GG and GA genotype of SNP rs1800801 (OR = 2.408, 95% CI 1.079–4.389, p = 0.004) were independently associated with recurrent stroke within the first year. A trend towards recurrent stroke was observed for positive family history (OR = 1.698, 95% CI 0.988–2.916, p = 0.055) (Table 3).

Subgroup analysis comparing the AA genotype vs GA genotype (excluding the GG genotype) within the same model showed an OR = 2.062, 95% CI 1.013–3.937, p = 0.029, thus demonstrating the AA genotype to be at highest risk. The area under the receiver operating characteristics curve (AUC) for the model including SNP rs1800801 was 0.744 (95% CI 0.717–0.770), whereas the model without the SNP rs1800801 yielded an AUC of 0.740 (95% CI 0.712–0.766) (Fig 1).

## Discussion

Identification of patients at risk for recurrent stroke is critical because approximately one-quarter of all strokes represent with recurrence and purport poor outcomes. Additional cerebrovascular ischemic events or systemic vascular ischemic events increase the risk for morbidity and mortality [1]. Thus, secondary prevention in patients who recently experienced a cerebrovascular event is critical and warrants investigation of clinical and non-clinical risk factors associated with stroke recurrence.

**Table 3. Predictors of recurrent stroke within first year after first stroke.**

| Predictors | ß coefficient | OR (95% CI) | P value |
|---|---|---|---|
| Procoagulant | 1.271 | 3.563 (1.504–8.443) | 0.004 |
| Prior stroke | 2.272 | 9.694 (5.793–16.224)) | ≤ 0.001 |
| Family history | 0.529 | 1.698 (0.988–2.916) | 0.055 |
| AA vs GG & GA genotype (rs1800801) | 0.879 | 2.408 (1.079–4.389) | 0.004 |

Subgroup analysis of AA vs GA genotype (rs1800801) in same multivariable model OR = 2.062, 95% CI (1.013–3.937, p = 0.029).

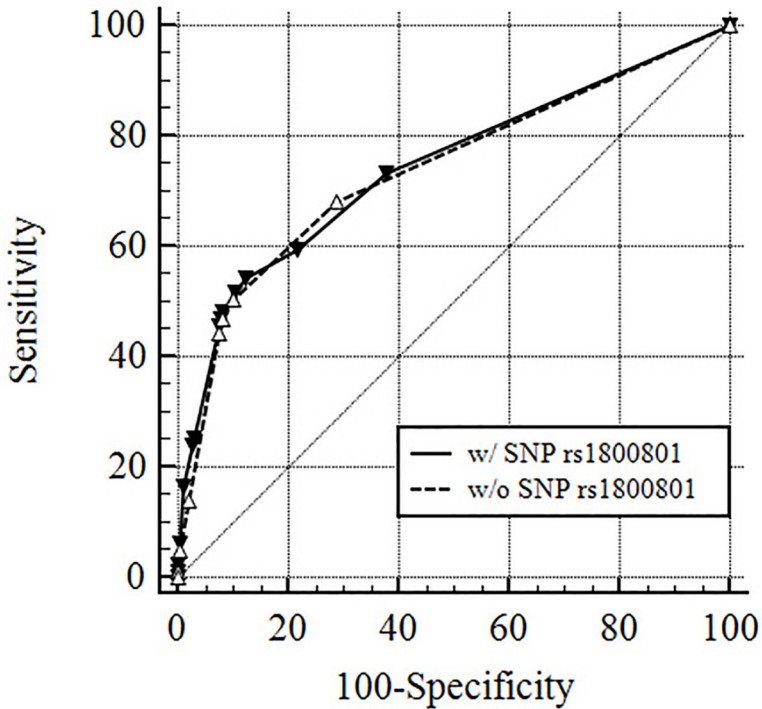

**Fig 1. Receiver operating curves for prediction of recurrent stroke within the first year with and without MGP single nucleotide polymorphism rs1800801.**

In this analysis, we observed an independent association between the AA genotype of the MGP SNP rs1800801 and recurrent stroke within the first year in North American Caucasians. The MGP is an extracellular matrix protein primarily synthesized by vascular smooth-muscle cells (VSMCs) and chondrocytes. Mice deficient in MGP display significant artery calcification and suffer pathological cartilage calcification causing osteopenia and fractures. [10]. The underlying mechanism of vascular calcification and the role of MGP in this multifactorial process are still under investigation. An interplay of uremic toxins, calcium and phosphate deposits, lipidaceous vesicles, and VSMC transdifferentiation has been debated [16]. Herrmann and colleagues identified 8 MGP polymorphisms and found that individuals with femoral atherosclerotic plaques calcifications and myocardial infarction were more frequently carriers of the minor A allele of rs1800801. However, they could not demonstrate a function for this allele in vitro [17]. Harbuzova et al. analyzed the role of MGP in cerebrovascular disease. They found that the AA genotype of rs1800801 was associated with an increased risk of ischemic atherothrombotic stroke in Ukrainian females [18,19]. They also reported that the AA genotype associated with an altered prothrombin time, presumably causing increased hypercoagulability [20]. This study group also observed the AA genotype of rs1800801 to occur significantly more frequently in patients that suffered from acute coronary syndrome [21]. Recently, Sheng et al. examined the association of the MGP SNPs rs1800801, rs1800802, and rs4236 with vascular calcification and atherosclerotic disease in a meta-analysis [22]. Out of 23 included case-control studies with 5,280 cases and 5,773 controls, the A allele of rs1800801 associated significantly with vascular calcification and atherosclerotic disease in Caucasians but not Asians. Moreover, no association was found with rs1800802 or rs4236. Fernández-Cadenas and colleagues recently reported that rs1800801 associated with first-year recurrent ischemic stroke in a Spanish cohort in the Genotyping Recurrence Risk of Stroke (GRECOS) study [9].

Interestingly, the G allele was the risk allele and the A allele was protective. Their derivation cohort consisted of 1,494 white patients enrolled in 23 Spanish hospitals and the results were replicated in a Spanish cohort of 1,305 patients. However, the authors did not find significant association of rs1800801 with recurrent ischemic stroke in a North American cohort of 1,683 patients who were also enrolled in the Vitamin Intervention for Stroke Prevention (VISP) trial [23]. The GRECOS authors raised the possibility that vitamin intervention in the VISP trial, genetic differences between North American and Spanish Caucasian populations, and/or exclusion criteria in the VISP trial could account for the discord between the Spanish and North American cohorts [9]. The findings in the current study of a Caucasian North American cohort are also discordant with those of Fernándes-Cadenas et al. We identified the allele A of rs1800801 as the risk allele. This finding is in line with the findings, discussed above, that the A allele is the risk allele in cerebrovascular disease.

Whether genetic information adds significant value to predict recurrent stroke has recently gained increasing attention. Achterberg et al. found that genetic information did not improve a risk stratification model for recurrence of vascular events including stroke after cerebral ischemia. The AUC was negligibly increased from 0.65 (95% CI 0.54–0.65) to 0.66 (95% CI 0.54–0.66) when adding genetic information [8]. In the current study, the predictive value of recurrent ischemic stroke was also not improved significantly by rs1800801 genetic information. The AUC of the prediction model when including the genetic status of rs1800801 was 0.744 as compared to 0.740 in the prediction model without inclusion of genetic status. Stroke recurrence is clearly multifactorial and the extent to which this or other SNPs or a combination thereof have an impact is currently unknown. Comparison of the two AUCs clearly illustrates the limitations of genetic factors when analyzed in combination with clinical characteristics. Thus, additional investigation is required to determine whether genetic differences, including MGP SNP rs1800801, contribute to the risk of recurrent stroke in acute ischemic stroke patients and whether they can improve accuracy when integrated into clinical prediction models. To date, the role of MGP SNP rs1800801 is somewhat inconclusive. With respect to prediction of recurrent ischemic stroke, it remains to be determined whether MGP SNP rs1800801 is a valid biomarker or surrogate marker.

## Limitations

Data collection and analysis were performed retrospectively and, as such, are subject to incomplete datasets. Association of a SNP does not provide insight into whether the given SNP is in linkage disequilibrium with another SNP(s) to account for the findings. We focused on rs1800801 and first-year recurrent risk of stroke based on the findings by Fernándes-Cadenas et al. [9]. A recurrent stroke within the first year was observed in 7.4% of our cohort. This is in line with other studies [2,9,24]. Whether setting the cutoff for early recurrence of stroke at a year is vindicated will require further exploration. The purpose of the presented study was to validate the work of Fernándes-Cadenas et al. where only recurrences within the first year were assessed [9]. The distribution of TOAST stroke subtypes and other clinical characteristics are comparable to other large stroke studies [15,25]. Additionally, the MyCode stroke database is not subject to selection bias [26]. Some patients may have had a recurrent stroke managed outside of our healthcare system during the follow-up period or suffered another stroke without further treatment.

## Conclusions

We found that the AA genotype of MGP SNP rs1800801 is associated with recurrent stroke within the first year after ischemic stroke in North American Caucasian. Whether

incorporation of the genetic status at this locus can contribute to a more accurate prediction of recurrent ischemic strokes in subgroups of Caucasians or other populations remains to be determined.

## Author Contributions

**Conceptualization:** Philipp Hendrix, Natalia S. Rost, James E. Faber, Christoph J. Griessenauer.

**Data curation:** Philipp Hendrix, Nelson Sofoluke, Matthew Adams, Saran Kunaprayoon, Ramin Zand, Amy N. Kolinovsky, Thomas N. Person, Mudit Gupta, Christoph J. Griessenauer.

**Formal analysis:** Philipp Hendrix, Amy N. Kolinovsky, Thomas N. Person, Mudit Gupta, H. Lester Kirchner, Christoph J. Griessenauer.

**Funding acquisition:** Christoph J. Griessenauer.

**Investigation:** Philipp Hendrix, Nelson Sofoluke, Matthew Adams, Saran Kunaprayoon, Ramin Zand, Amy N. Kolinovsky, Mudit Gupta, H. Lester Kirchner, Clemens M. Schirmer, Natalia S. Rost, Christoph J. Griessenauer.

**Methodology:** Philipp Hendrix, Thomas N. Person, H. Lester Kirchner, Natalia S. Rost, James E. Faber, Christoph J. Griessenauer.

**Project administration:** Philipp Hendrix, Ramin Zand, Oded Goren, Clemens M. Schirmer, Christoph J. Griessenauer.

**Resources:** Philipp Hendrix, Christoph J. Griessenauer.

**Software:** Philipp Hendrix, H. Lester Kirchner.

**Supervision:** Philipp Hendrix, Oded Goren, H. Lester Kirchner, Clemens M. Schirmer, Natalia S. Rost, Christoph J. Griessenauer.

**Validation:** Philipp Hendrix, H. Lester Kirchner, Clemens M. Schirmer, Christoph J. Griessenauer.

**Visualization:** Philipp Hendrix, Christoph J. Griessenauer.

**Writing – original draft:** Philipp Hendrix, James E. Faber, Christoph J. Griessenauer.

**Writing – review & editing:** Philipp Hendrix, Ramin Zand, Oded Goren, H. Lester Kirchner, Clemens M. Schirmer, Natalia S. Rost, James E. Faber, Christoph J. Griessenauer.

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
