## [Decision Letter · Decision Letter 0]

29 May 2020

PONE-D-20-07538

Matrix Gla protein polymorphism rs1800801 associates with recurrence of ischemic stroke

PLOS ONE

Dear Dr. Hendrix,

Thank you for submitting your manuscript to PLOS ONE. After careful consideration, we feel that it has merit but does not fully meet PLOS ONE’s publication criteria as it currently stands. Therefore, we invite you to submit a revised version of the manuscript that addresses the points raised during the review process.

We look forward to receiving your revised manuscript.

Kind regards,

Aurel Popa-Wagner

Academic Editor

PLOS ONE

Journal Requirements:

'The authors received no specific funding for this work.'

We note that one or more of the authors are employed by a commercial company: Geisinger Health System

3. We note you have included a table to which you do not refer in the text of your manuscript. Please ensure that you refer to Table 3 in your text; if accepted, production will need this reference to link the reader to the Table.

Reviewers' comments:

Reviewer's Responses to Questions

**Comments to the Author**

1. Is the manuscript technically sound, and do the data support the conclusions?

Reviewer #1: Yes

Reviewer #2: Partly

2. Has the statistical analysis been performed appropriately and rigorously? 

Reviewer #1: Yes

Reviewer #2: Yes

3. Have the authors made all data underlying the findings in their manuscript fully available?

Reviewer #1: Yes

Reviewer #2: Yes

4. Is the manuscript presented in an intelligible fashion and written in standard English?

Reviewer #1: Yes

Reviewer #2: Yes

5. Review Comments to the Author

Reviewer #1: In the Genotyping Recurrence Risk of Stroke (GRECOS) project, however, Fernándes-Cadenas et al. found an association of the single nucleotide polymorphism (SNP) rs1800801 in the matrix carboxyglutamatic acid Gla protein (MGP) gene with first-year recurrent ischemic stroke in Spanish Caucasians. MGP is an

extracellular matrix protein involved in inhibition of bone and vascular calcification. Since a recent a meta-analysis highlighted the SNP rs1800801 (12-15038788C-T, G>A) in Caucasians to increase the risk for vascular calcification and atherosclerotic disease, the authors investigated the impact of rs18008001 on the risk for first-year recurrent stroke in a North American Caucasian population. This is a good study conducted on a acceptable no of patients. However, since age is the only non-modifiable risk factor for stroke I would like to know the age-dependence (if any) of the SNP rs1800801 among the enrolled patients. Further, the authors shall discuss the age-dependency of extracellular matrix protein involved in inhibition of bone and vascular calcification.

Reviewer #2: -what is the normal distribution of the AA genotype of MGP SNP rs1800801 in general healthy Caucasian population of North America? Is it comparable with the population of Spain?

-in Table 1 – Patients characteristics - in the stroke recurrent group: 44.3% had stroke in the past (vs. 7.6%), which may be in relation with genetic association (family history), associated with metabolic syndrome (80% with BMI>25, in both groups), and procoagulant disorders. Also it seem that the recurrence appear in nonsmokers group(43% vs. 25.3%). How would you comment on the lack of correlation between smoking and vascular complications? Studies did show an increase of MGP in smokers (Silaghi CN et al, Clin Chim Acta 2019) and the correlation with cardiovascular complications.

- how would you comment on the fact that mortality was less probable in the recurrent group (even with prior strokes) than in the other group (1.26% vs. 2.43%).

- why should be taken in consideration the AA genotype frequency of SNP rs1800801, representing 13% of genotypes (vs.39.5% and 47.4%), from 7.4% of patients that will have recurrence in the first year, and these events are less harmful than in the other group? How powerful is the message for introducing AA genotype of SNP rs1800801 as a risk tool, in comparison with other contributing factors to vascular complications? If other risk factors are not described, they should be at least mentioned – SNP for adiponectin, etc.

-rows 70-71, 190 – „MGP is an extracellular matrix protein involved in inhibition of bone and vascular calcification” - is misleading information. MGP is not produced by bone cells, but by chondrocytes and fibroblasts, smooth muscle cells (Price PA, 1988, 1992). MGP does not have a role in bone remodeling cycle.

6. PLOS authors have the option to publish the peer review history of their article (what does this mean?). If published, this will include your full peer review and any attached files.

Reviewer #1: No

Reviewer #2: No

---

## [Author Response · Author response to Decision Letter 0]

7 Jun 2020

Reviewer #1: In the Genotyping Recurrence Risk of Stroke (GRECOS) project, however, Fernándes-Cadenas et al. found an association of the single nucleotide polymorphism (SNP) rs1800801 in the matrix carboxyglutamatic acid Gla protein (MGP) gene with first-year recurrent ischemic stroke in Spanish Caucasians. MGP is an

extracellular matrix protein involved in inhibition of bone and vascular calcification. Since a recent a meta-analysis highlighted the SNP rs1800801 (12-15038788C-T, G>A) in Caucasians to increase the risk for vascular calcification and atherosclerotic disease, the authors investigated the impact of rs18008001 on the risk for first-year recurrent stroke in a North American Caucasian population. This is a good study conducted on an acceptable no of patients. 

We appreciate the reviewer´s positive comment on our study. 

However, since age is the only non-modifiable risk factor for stroke I would like to know the age-dependence (if any) of the SNP rs1800801 among the enrolled patients. 

We appreciate the reviewer´s comment. We assessed age-dependent distribution of SNP rs1800801 genotypes in our cohort. We did not find age differences among AA, GA, GG genotypes in our cohort (mean 69 – 68 – 67; median 70 – 70 – 68; min/max/IQR 24/93/17 – 22/96/20 – 28/99/19, respectively). We also assessed age groups and dichotomized for cut-off values (age ≥ 60 as in ABCD2/California scores or age ≥ 69 as present study median) and did not find any differences. In our logistic regression model, impact of age among enrolled patients was assessed both as a quantitative (continuous) variable and a qualitative (age groups) variable. We agree that age is an important risk factor for stroke recurrence, which is also displayed by ABCD2, ESRS, and California risk score to predict recurrent events. However, the herein presented study is a retrospective database analysis, which should not compete with prospective clinical risk stratification. 

Further, the authors shall discuss the age-dependency of extracellular matrix protein involved in inhibition of bone and vascular calcification.

The reviewer addresses an interesting issue. To date, the mechanisms of vascular calcification and the interplay of MGP have not been elucidated. We added a statement on the potential players in the pathological mechanisms. Elaborating on unsupported hypotheses, however, would side track the readers´ attention from the genetic association study. Thus, we limited the additional information on this aspect.

Reviewer #2: -what is the normal distribution of the AA genotype of MGP SNP rs1800801 in general healthy Caucasian population of North America? Is it comparable with the population of Spain?

The references (ref) and alternative (alt) allele frequencies can be obtained from reference [14] in the manuscript: https://www.ncbi.nlm.nih.gov/projects/SNP/snp_ref.cgi?rs=1800801. The minor allele frequencies (MAFs) appear to be similar for most ethnicities and range from 0.27 – 0.37, whereas a significant decline has been observed in some Asian subgroups with MAF < 0.07. Therefore, we expect the MAF from our North American cohort (only Caucasians) and the GRECOS cohort to be in equal ranges.

-in Table 1 – Patients characteristics - in the stroke recurrent group: 44.3% had stroke in the past (vs. 7.6%), which may be in relation with genetic association (family history), associated with metabolic syndrome (80% with BMI>25, in both groups), and procoagulant disorders. Also it seem that the recurrence appear in nonsmokers group (43% vs. 25.3%). How would you comment on the lack of correlation between smoking and vascular complications? 

Smoking status between the two groups (1st-year recurrent vs no recurrent) was equally distributed as displayed in Table 1. In the 1st-year recurrent group 25.3% were current smokers, whereas 43.0% had never smoked. In the no recurrent group 21.2% were current smokers and 38.5% had never smoked. We cannot replicate that non-smoking should be the predominant driver for stroke recurrence.

Additionally, we do not report a lack of correlation between smoking and vascular complications; as such analysis is not part of the herein presented manuscript. In our study, smoking status was not associated with 1st-year stroke recurrence. However, in our cohort we do find that never-smokers have significant decreased odds of suffering from carotid stenosis, coronary heart disease, and peripheral vascular disease (all OR < 1, p < 0.05). That smoking status was not associated with 1st-year stroke recurrence could be attributed to multiple factor: i) data was obtained from a retrospective database, thus smoking status could not be analyzed in a quantitative but only a qualitative measure, ii) effects of former-smoking could counterbalance both current and never smoking depending on how long smoking has been stopped, or iii) smoking might not be a predominant factor for early stroke recurrence. Additional studies need to investigate the role of smoking for early stroke recurrence.

Studies did show an increase of MGP in smokers (Silaghi CN et al, Clin Chim Acta 2019) and the correlation with cardiovascular complications.

- how would you comment on the fact that mortality was less probable in the recurrent group (even with prior strokes) than in the other group (1.26% vs. 2.43%).

The 1.3% vs 2.4% mortality rates are those at patient discharge. Discharge outcomes rates are prone to bias. Duration of hospital stay, complications during hospital stay, and discharge location have to be taken into consideration when interpreting discharge outcomes. Therefore, standardized follow-ups such as 90-day follow-ups are mandatory to evaluate outcome. Here, at 90-day follow-up, mortality rates are 7.8% vs 5.9% which reflects the burden of a recurrent stroke event.

- why should be taken in consideration the AA genotype frequency of SNP rs1800801, representing 13% of genotypes (vs.39.5% and 47.4%), from 7.4% of patients that will have recurrence in the first year, and these events are less harmful than in the other group? 

As mentioned above, recurrent strokes are not less harmful event that no recurrent strokes. The AA genotype was significantly associated with occurrence of a first-year stroke recurrence. 

How powerful is the message for introducing AA genotype of SNP rs1800801 as a risk tool, in comparison with other contributing factors to vascular complications? If other risk factors are not described, they should be at least mentioned – SNP for adiponectin, etc.

In the results section we demonstrate that we could not improve predictability of stroke recurrence with the integration of SNP rs1800801, i.e. the AUC are almost identical. In the discussion section we mention that we did not find this genetic polymorphism to be of additional predictive value and thus far clinical variables appear to be most accurate. 

-rows 70-71, 190 – „MGP is an extracellular matrix protein involved in inhibition of bone and vascular calcification” - is misleading information. MGP is not produced by bone cells, but by chondrocytes and fibroblasts, smooth muscle cells (Price PA, 1988, 1992). MGP does not have a role in bone remodeling cycle.

We agree with the reviewer that the abstract from our reference reports vascular smooth muscle cell and chondrocyte calcification. We do not mention that MGP is synthesized by osteoblasts. However, a detailed look into the reports also mentioned by the reviewer (Fraser and Price 1988, J Biochem) reveals: “It should be noted in this context that MGP is in fact made by cartilage as well as bone.“ Additionally, Cancela and Price 2001: Matrix Gla protein in Xenopus laevis: molecular cloning, tissue distribution, and evolutionary considerations. J. Bone Miner. Res. extract MGP from bones and the role of MGP in bone remodeling is under investigation. However, this is beyond the scope of the manuscript.

Our cited reference refers to enchondral ossification in mice and thus the term “bone calcification” is likely not precise. We corrected this according

---

## [Editor Report · Decision Letter 1]

10 Jun 2020

Matrix Gla protein polymorphism rs1800801 associates with recurrence of ischemic stroke

PONE-D-20-07538R1

Dear Dr. Griessenauer,

We’re pleased to inform you that your manuscript has been judged scientifically suitable for publication and will be formally accepted for publication once it meets all outstanding technical requirements.

Kind regards,

Aurel Popa-Wagner

Academic Editor

PLOS ONE
---

## [Editor Report · Acceptance letter]

16 Jun 2020

PONE-D-20-07538R1 

Matrix Gla protein polymorphism rs1800801 associates with recurrence of ischemic stroke 

Dear Dr. Griessenauer:

I'm pleased to inform you that your manuscript has been deemed suitable for publication in PLOS ONE. Congratulations! Your manuscript is now with our production department. 

Kind regards, 

on behalf of

Professor Aurel Popa-Wagner 

Academic Editor

PLOS ONE